# In-Situ Forming pH and Thermosensitive Injectable Hydrogels to Stimulate Angiogenesis: Potential Candidates for Fast Bone Regeneration Applications

**DOI:** 10.3390/ijms21051633

**Published:** 2020-02-27

**Authors:** Fatma Z. Kocak, Abdullah C.S. Talari, Muhammad Yar, Ihtesham U. Rehman

**Affiliations:** 1Engineering Department, Lancaster University, Lancaster LA1 4YW, UK; f.kocak@lancaster.ac.uk (F.Z.K.);; 2Interdisciplinary Research Centre in Biomedical Materials (IRCBM), COMSATS University Islamabad, Lahore Campus, Punjab 54000, Pakistan; drmyar@cuilahore.edu.pk

**Keywords:** thermosensitive injectable hydrogels, injectability, angiogenesis, ex-ovo chick chorioallantoic membrane (CAM) assay, chitosan, hydroxyapatite, heparin

## Abstract

Biomaterials that promote angiogenesis are required for repair and regeneration of bone. In-situ formed injectable hydrogels functionalised with bioactive agents, facilitating angiogenesis have high demand for bone regeneration. In this study, pH and thermosensitive hydrogels based on chitosan (CS) and hydroxyapatite (HA) composite materials loaded with heparin (Hep) were investigated for their pro-angiogenic potential. Hydrogel formulations with varying Hep concentrations were prepared by sol–gel technique for these homogeneous solutions were neutralised with sodium bicarbonate (NaHCO_3_) at 4 °C. Solutions (CS/HA/Hep) constituted hydrogels setting at 37 °C which was initiated from surface in 5–10 minutes. Hydrogels were characterised by performing injectability, gelation, rheology, morphology, chemical and biological analyses. Hydrogel solutions facilitated manual dropwise injection from 21 Gauge which is highly used for orthopaedic and dental administrations, and the maximum injection force measured through 19 G needle (17.191 ± 2.296N) was convenient for manual injections. Angiogenesis tests were performed by an ex-ovo chick chorioallantoic membrane (CAM) assay by applying injectable solutions on CAM, which produced in situ hydrogels. Hydrogels induced microvascularity in CAM assay this was confirmed by histology analyses. Hydrogels with lower concentration of Hep showed more efficiency in pro-angiogenic response. Thereof, novel injectable hydrogels inducing angiogenesis (CS/HA/Hep) are potential candidates for bone regeneration and drug delivery applications.

## 1. Introduction

The conventional cell-scaffolds techniques in tissue engineering require direct surgical operations leading causality of bones, devastation of vicinal tissues and traumas as well as difficulties in their production and cell seeding into their inner cavities. Hereof, in situ form-injectable hydrogels have drawn attentions since they offer minimal invasive straightforward injection and subsequent crosslinked gel formation by moulding complex tissue defects in the body stimulated by an external or internal stimulus [1,2]. Hydrogels as extra cellular matrix (ECM) mimetic materials due to their highly hydrated, permeable and porous structures enable guided tissue regeneration facilitating cell activities, nutrition and waste transfer through their crosslinked networks [3]. Swelling and degradation ability of polymer matrix hydrogels make them suitable vehicles to encapsulate and deliver numerous therapeutic agents, such as cells, growth factors, drugs, and genes at tissue defects [4].

In-situ forming thermosensitive hydrogel found as liquid in room temperature and form gel via an internal stimuli of natural body temperature. Thermosensitive polymer hydrogels undergo a phase alteration owing to presence of similar amount of hydrophobic and hydrophilic groups in their networks [5]. Mostly, thermally responsive hydrogels are soluble below a certain temperature, which is known as lower critical solution temperature (LCST) whereas they form insoluble gels above this temperature in body [6]. 

On the other hand, pH responsive gelation occurs due to the pH related ionic interactions leading proton loose or gain. The swelling in pH sensitive hydrogels is monitored by alteration of electrostatic charge of the ionised pendant sites of polymers leading electrostatic repulsion. pH sensitive hydrogels having anionic pendent groups e.g., carboxylic and sulphated groups are entitled as anionic hydrogels whereas cationic pendent groups (for instance, amine) involved hydrogels are designated as cationic hydrogels. Anionic hydrogels (such as carboxymethyl chitosan) swell when pH is raised above pK_a_, which is feasible for drug delivery at high pH (e.g., intestine possess pH 7.4). On the other hand, cationic hydrogels (such as chitosan) swell in low pH under the pK_b_ once ionisation occur, and beneficial for drug delivery in acidic pH e.g., in stomach [7,8]. 

Drug delivery via diffusion of particles from hydrogel network was based on Fick’s Law [8,9]. The diffusion of molecules depends on net ionic charge in multi-ion systems. For instance, in counter charged polycomplex systems, particle diffusion is retarded due to electrostatic bonding. Solute diffusion and partitioning in hydrogels have been correlated by ionic, chemical and steric interactivity and concentration in theories based on Poisson–Boltzmann (PB) cell model [9,10]. The partition coefficient equality is given at the Equation (1) [9].
(1)Φ=θαπ=[A]g/[A]w
(Φ: partition coefficient, θ, α and π: steric, chemical and electrostatic interactions, and [A]_g_ and [A]_w_: concentration of solute in gel and water, respectively.)

Among in-situ formed hydrogels, pH and thermosensitive hydrogels are more attractive due to the absence of any toxic cross-linkers or external stimuli, such as UV and their possible adverse effects against surrounding cells and ECM by elevated temperature during polymer formation in-situ [11]. Since pH and thermosensitive hydrogels are physically crosslinked, they provide a temporary mechanical support for non-load bearing bones. They stimulate the cells for regrowth of tissues with the active agents introduced to the ECM through the swelling and degradation of hydrogels. Once the regeneration of injured bones start, polymer matrixed hydrogels start to degrade [3,12].

A wider range of different thermoresponsive gels have been used for biomedical applications in recent years. One commonly used thermosensitive hydrogel in tissue engineering is Poly(*N*-isopropylacrylamide) (PNIPAAm) forming nearly at 30 °C. The elastic PNIPAAm hydrogels integrated with cardiac stem cells for hearth regeneration to stimulate cardiac differentiation [13] and PNIPAAm crosslinked with hyaluronic acid forming injectable porous hydrogels which was biocompatible with encapsulated adipose derived stem cells have been reported [14]. Recently, highly tough hydrogels from PNIPAAm and alginate composites whose surfaces were modified by chitosan and carbodiimide crosslinker for adhesive and contacting via body temperature which accelerate wound healing in-vivo in mice has been reported. Another example of thermosensitive hydrogels are Pluronics are polyethylene oxide-based triblock copolymers, known for their bioadhesion, stability and thermosensitive gel forming capability at 30 °C, and are commonly used for injectable drug delivery systems [15]. In addition, polyurethane based hydrogels constituted by poly(glycerol-sebacate) and poly(ethylene-glycol) having thermo-reversibility between 5 and 37 °C has been reported. These hydrogels showed elastic, biodegradable and biocompatible features with adipose derived cells for soft tissue engineering applications [16]. Poly(*N*-vinylcaprolactam), (PNVCL) based nanogels have shown antiviral activity against HIV-virus [17] whereas PNVCL–PEGMA based nanogels have shown great potential in 5-Fluorouracil (5FU) drug delivery systems [18]. Poly(VCL-co-UA), Poly(vinylcaprolactam) (PVCL) and Poly(*N*-isopropylacrylamide) (PNIPAM)-based microgels have been promising candidates in controlled drug delivery of anticancer drugs, such as, doxorubicin [19,20,21]. A combination of nanocomposite hydrogels in glycerol–water cosolvent (GW) gels have great potential to be used in biosensing, soft-robotics and artificial intelligence due to their electrical sensitivity towards environmental temperature changes [22,23]. A combination of tranexamic acid (TXA) and thermosensitive CS-β-GP (β-glycerophosphate) gel/spray has promoted nasal wound healing in the treatment of epistaxis [24]. A sol–gel fabrication of antibacterial peptide nanofibrils with PNIPAM have shown great promise for minimally invasive drug delivery [25]. A combination of PNIPAM in hybrid PVA and sodium tetraborate decahydrate composite hydrogels have potential in the development of artificial electronic skins and temperature dependent soft electronics applications [26]. Sertaconazole-loaded nanostructured lipid carriers (Sertaconazole-NLCs) incorporated HPMC nanocomposite hydrogels have shown good results in vitro studies in the treatment of fungal keratitis [27]. In-vitro studies have revealed that silver sulfadiazine (AgSD)/NsS loaded thermosensitive gels exhibited bactericidal effect on pathogens during wound healing process [28]. PLGA-PEG (PP) copolymer with simvastatin (SIM) nanoparticles combined with porous bioceramic scaffolds have given promising results in bone regeneration in radial bone defect in rabbits [29].

Chitosan being a natural cationic long linear chain polysaccharide able to form pH and thermosensitive hydrogels at the vicinity of body temperature [12]. Chitosan based biomaterials involving hydrogels have been used in diverse tissue engineering applications. Chitosan based composite hydrogels and films harnessed as wound healing material with drug delivery [30]. Chitosan gel-scaffolds combined with pectin or hyaluronate have been explored in tooth root canal regeneration in dog teeth [31]. In addition, chitosan gels and its combination with collagen or demineralized bone clinically have been utilised for regeneration of periodontal bone defects in humans. The significant bone mineralisation obtained in all groups as the best bone filling achieved with the collagen and chitosan compositions [32].

Chitosan based thermosensitive injectable hydrogels involving pH neutralisation by using β-glycerophosphate disodium β-(GP) and gelation at the vicinity of 37 °C driven by intra-hydrogen binding of chitosan due to decrease in repulsive forces via basicity of salt and further hydrophobic interactions due to the increment at the temperature [33]. BTS Car-Gel^®^ commercial thermosensitive chitosan solution neutralised by β-(GP) have been utilised to treat cartilage lesions after trauma or joints via bone marrow stimulation method in clinical trials. BTS Car-Gel solution mixed with blood was applied arthroscopic or minimal surgery methods stabilised blood clotting and adhered the defects and induced cartilage repair significantly [34,35]. 

Although the beneficial application of β-(GP) neutralised hydrogels, to achieve a physiological pH level, high dosages of β-(GP) is required mitigating cell cytocompatibility [36]. Therefore, some researchers have exploited different neutralising agents as additive to β-GP in thermosensitive chitosan hydrogels, such as Mg GP to investigate cell adhesion and proliferation via in-vitro osteosarcoma cells [37] and Ca GP to investigate mineralisation to ensure calcium phosphate mineralisation [38]. Furthermore, di-sodium carbonate (Na_2_CO_3_) [39], and sodium bi-carbonate (NaHCO_3_) [40] compounds have been investigated as pH and temperature sensitive hydrogels. 

In pH and temperature sensitive chitosan solutions neutralised by a weak base NaHCO_3_, gelation occurs gradually from the surface to bulk. The gelation is driven by protonation and deprotonation of amine groups in chitosan reversibly reacting with HCO_3_^−^ in NaHCO_3_ and H^+^ ions in the acidic environment provide an ionic crosslinking of chitosan with neutral NH_2_ linkages. The accompanying reaction product, carbon dioxide can metabolically release from the body which is non-toxic. By preventing carbondioxide release, stability of solution is increased for in-situ gelation which is also stimulated by body temperature [41]. The harnessing sodium bicarbonate addition to β-GP have provided increase in cell cytocompatibility, as well as enhancing mechanical and gelation properties [42]. 

Chitosan and hydroxyapatite incorporation in biomaterials due to inherited benefits of hydroxyapatite for bone regeneration have been investigated widely. The impact of hydroxyapatite on bone regeneration can be associated with its origin comprising the inorganic matrix of natural bones with diverse ionic substitutions in its crystal lattice. The unique bioactive properties of hydroxyapatite promotes cell adhesion and osteoconductivity, as well it enhances proliferation and alkaline phosphate secretion of osteoblasts, which helps early healing of bones [43,44,45]. Therefore, hydroxyapatite has significant contributions in bone regeneration involving osteogenic response [44], biocompatibility and osteoblast attachment [46], in vivo–bone regrowth in rats [47] periodontal tissue regeneration with anti-inflammatory potential [48], proliferation and differentiation of alveolar bone regeneration accompanied with drug release [49].

Recently, chitosan/hydroxyapatite (CS/HA) thermosensitive hydrogels neutralised by β-GP has been reported [50] by utilisation of water soluble thiolated chitosan for bovine serum albumin release. In addition, CS/HA thermosensitive hydrogels with gentamicin were harnessed to increase porosity of bone cement PMMA, this led to enhancement of eventual features of composite with better mineralisation and cell viability [51]. In another study, anti-inflammatory hydrogels from CS, HA and poly(vinyl-alcohol) system has investigated for drug (meloxicam) delivery for periodontal treatments. In addition to swelling and sustainable drug release, these hydrogels were cytocompatible and improved cell proliferation with anti-inflammatory properties [48]. Recently, pH and thermosensitive chitosan/hydroxyapatite hydrogels neutralised by sodium bicarbonate was integrated by in-situ formed hydroxyapatite have contributed to increment in pH for neutralisation, and good cell viability and proliferation was obtained in encapsulated cells [40].

Minimal invasive hybrid hydrogels functionalised with bioactive agents not only facilitating interfacial bonding with natural bones but also providing vascular network bridging via porous stimulation of angiogenesis by active agents are crucial for eventual bone regeneration. Therefore, in this study, pH and thermosensitive chitosan based injectable hydrogels were investigated for functional guided bone tissue regeneration by stimulating angiogenesis via recruiting physiologically active glycosaminoglycan, heparin.

As has been reported the most failure in the regeneration and repair of bone tissues root from insufficient vascularisation [52,53]. Once tissue engineered biomaterials are implanted into large-scaled bone defects, the tissue healing strongly rely on keeping cells alive by swift branching of present blood vessels occupying defect sites. In case of deficient blood supply, cells cannot survive since their metabolic activities are impaired because of supplement of nourishments, oxygen, and transfer of metabolism products. The sufficiently grown microvascular structure also maintains the regulation of various cells which participate in remodelling and reconstruction of tissues, including hematopoietic stem cells, osteoprogenitor and immune cells [54]. Therefore, there is surplus demand of pro-angiogenic biomaterials to stimulate regeneration of tissues via angiogenesis. 

To stimulate angiogenesis for tissue healing, one common strategy is delivery of pro-angiogenic growth factors, such as vascular endothelial growth factor (VEGF), basic fibroblast growth factor (bFGF), and transforming growth factor beta (TGFβ) [55,56,57]. Although they substantially stimulate angiogenesis, their efficiency is restricted by their instability due to high solubility that reduce their performance and activation time. This becomes more challenging upon fracture in tough conditions. To combat their instability, the approach to utilise high level of growth factors also leads toxic effects and malformation of bones. In addition, regional inflammatory reactions, adverse effects on nerves and risk of malignant tissue formations have been reported [58,59].

Due to instability issues of growth factors, they are mostly harnessed by their immobilisation into a scaffold matrix providing their controlled delivery. One approach is to bind these growth factors with heparin or heparan sulfate proteoglycans (HPSGs) since they have great binding capacity of proteins involving growth factors due to their highly anionic nature [60]. Heparin and HPSGs are proficient to bind pro-angiogenic receptors with majority of angiogenic growth factors, comprising VEGF, bFGF, and TGFβ [56,61]. The binding heparin with pro-angiogenic factors provides proliferation, generation of protease and migration of endothelial cells, as well as leading cells for blood vessel formation [56].

Recently in our group, heparin was exploited to induce angiogenesis investigated via CAM assay to evaluate the potential attachment of physiologically available angiogenic growth factors to pro-angiogenic receptors by using heparin bonded chemically crosslinked chitosan poly-vinyl alcohol (PVA) hydrogels. Triethyl orthoformate (TEOF) crosslinked and heparin bonded hydrogels led to more blood vessel generation as compared to heparin-free control samples [62]. In addition, direct mixing of heparin in chitosan-PVA-PCL hydrogels in the absence of any growth factors was investigated for angiogenesis on chick embryo’s CAM tissues for wound healing application. It was found that heparin bonded chitosan-PVA-PCL hydrogels led to significantly more angiogenesis than the sole collagen control gels [63]. 

To our knowledge, heparin inclusion (by blending in solution) in the injectable pH and thermosensitive chitosan-hydroxyapatite hydrogel systems neutralised by NaHCO_3_ to investigate their effects on angiogenesis for bone regeneration has not been investigated till date. Therefore, in this study, thermosensitive injectable CS/HA/Hep hydrogels have been developed in different compositions via optimisation of synthesis parameters for homogeneous sol formulations and hydrogels. These hydrogel compositions were thoroughly characterised comparatively for various aspects. Angiogenesis analyses were performed via an ex-ovo CAM assay in which in-situ gelation of thermosensitive hydrogel solutions applied on CAM tissue occurred upon incubation at 37.5 °C. This technique mimics the in-vivo sol–state application of injectable hydrogels, and the rapid gel formation has eliminated the involvement of using an external material to keep solution in a place on CAM such as, such as silicone ring which might cause irrelevant material–tissue reactions. 

The results have shown that the composite hydrogels with (CS/HA/Hep) and without heparin (CS/HA) have indicated pro-angiogenic response in CAM. However, it was found that the lowest concentration of heparin used was sufficient for angiogenesis while less angiogenesis was obtained at higher heparin concentrations compared to control samples (CS/HA). In addition, the highest concentration of Hep in hydrogels, as well as decreasing gelation temperature, has led to significant increase in elastic modulus in comparison to other compositions. This can be the indicator of strengthening of ionic bonding due to counter charged CS and Hep polycomplex formation. This could be useful for sustainable drug delivery as long as injectability is maintained for drug encapsulation in-situ for injectable drug administration.

## 2. Results

### 2.1. Synthesis of Hydrogels 

All hydrogel compositions have showed a proper flow through syringes coupled with 21 gauge and thicker size needles. The initial gelation of solutions has started from surface towards to the bulk in 5–10 minutes at 37 °C and the complete gelation occurred in 24–48 h in close-lid glass petri dishes or test tubes in 37 °C oven. 

Homogeneous chitosan and composite solutions were obtained at pH between 6.2–6.3 upon neutralisation with NaHCO_3_ while higher pH led to precipitation problems. The precipitation problem with addition of heparin in powder form was overcome by gradual addition of heparin as a solution in de-ionised water. Diverse hydrogel compositions with their codes and pH of homogeneous mixtures after neutralisation and gelation, and incipient gelation time in a silicone oil bath at 37 °C are summarized in Table 1.

Transparent yellow colour, homogeneous sole chitosan hydrogel solution (CI) and hydrogels were obtained after incubation at 37 °C and are shown in Figure 1. Sol–gel transition of thermosensitive hydrogel solutions occurred in minutes in test tubes placed in an oil bath at 37 °C which ranges from 5 to 10 min. Figure 2 shows test tube invert method to incipient gelation occurring at a composite hydrogel solution (CS/HA/Hep) in 7 min. Since the gelation is driven by gradual carbon dioxide evaporation from surface, the exact gelation time is found to be depend on the surface area and the depth of the solutions. In addition, it has been found that gelation of solutions occurred very fast in open-lid glass petri dishes while homogeneous gels were obtained in closed lid vials or plates gradually. To obtain homogeneous hydrogels, solutions were poured into one-level open a glass-tube (75 mm × 18 mm) with three level push polythene plug cap) and left in the oven at 37 °C until complete gelation for 2 mL of solution for 48 h.

#### 2.1.1. Injectability 

##### Qualitative Evaluation of Injectability

The hydrogel solutions with higher chitosan concentration (7 and 8 *w*/*v* in initial solution) were not able to pass through 20 G needles. They had already gel like viscosity in solution state. The formulations with lower concentrations 5 and 6%(*w*/*v*) have resulted in injectable (drop flow), transparent yellow solution (Figure 1). In addition, the incipient gelation occurred in test tube in 8 min with hydrogels involving 5% (*w*/*v*) initial chitosan solution concentration. Therefore, in the composite hydrogels, 5% (*w*/*v*) chitosan solution was used for optimal injectability and gelation features. 

The injectability rating of all hydrogel solutions are summarised in the Table 2. All solution compositions have shown perfect flow from 20 G and thicker sized syringes. Sample-III with the highest concentration of heparin was more viscous and had less flow rate from 22 G needle compared to other compositions. This composition had lower incipient gelation temperature as well in the rheology measurements as can be seen in the Section 2.3.

##### Quantitative Injectability Measurements

Injectability results were represented as a graph of force versus displacement measurements (Figure 3.). Initial glide force (IGF), dynamic glide force (DGF) and maximum force (F_max_) were counted and are given in Table 3. 

The F_max_ value was the highest in CI solution as compared to composite hydrogel solutions in both experiments with and without needle. DGF and F_max_ forces were similar in both CII and SI solutions while these forces for SI solution was slightly higher than CII solutions at experiments performed with 19 G needle.

The injectability through needleless syringe, F_max_ values were the same with the IGF while F_max_ were at least two times higher than IGF values at tests through a 19 G needle (BD, 19 G × 1 1/2” 1 × 40 mm). The DGF values at injection through 19 G needle were at least 10 times higher than those of needless syringe flow. However, all forces applied for injectability via a 19 G needle reached to maximum was about 17 N.

### 2.2. Solution Stability 

Synthesised thermosensitive solutions were stored in closed lid universal tubes in a fridge at 4 °C. Hydrogel solutions without heparin (CI and CII) persisted their fluid form up to six months. However, heparin involved composite hydrogels were maintained their stability as fluid up to three weeks.

### 2.3. Rheology Measurements 

A temperature sweep test between 15 to 60 °C was performed with a chitosan hydrogel solution (the initial CS concentration of 6% (*w*/*v*)) presented a sol–gel transition behaviour occurring at 35.9 °C (Figure 4). The elastic modulus has increased up to 496 Pa while the temperature was increasing until 60 °C. The high elastic modulus of hydrogel was also maintained during the cooling temperature down back to 15 °C. This has revealed thermo-irreversible gel formation behaviour which was obtained in all compositions. There was a slight decrease in both viscous (G″) and elastic modulus (G′) from 15 to nearly 30 °C then, elastic modulus started to increase gradually.

Time sweep tests in which the temperature was kept at 37 °C for 30 min have shown that elastic modulus has gradually increased with time while viscous modulus stayed constant. This is the indicator of the mechanical strengthening of gels in body temperature with the time. As it can be seen in the Figure 5, the elastic modulus of chitosan hydrogel has increased from 252 Pa to 619 Pa during 30 min time sweep.

In Figure 6, there is a comparative rheology measurement of two control samples (CI and CII) and samples with heparin (SI and SIII). The CI sample with only chitosan had an incipient gelation at almost 37 °C (Figure 6a). Hydroxyapatite addition into hydrogels has contributed to decrease the gelation temperature to almost 31 °C (Figure 6b). Furthermore, heparin has further impact on reducing the incipient gelation temperature to approximately 28 and 21 °C in SI and SIII samples, respectively (Figure 6c,d). Moreover, the highest concentration of heparin in SIII significantly enhanced elastic modulus of hydrogels from approximately 5 Pa up to 130 Pa (Figure 6d). However, these more viscous formulations had quicker gelation during storage at 4 °C which limits the injectability and storage.

### 2.4. Scanning Electron Microscopy (SEM) 

The SEM images of freeze-dried hydrogel foams revealed an interconnected porous morphology (Figure 7). The mean pore diameter counted in Image-J^®^ (Version 1.52 k, National Institutes of Health, Bethesda, MD, USA) software was 73.4 µm, while minimum and maximum length of pores were 19.5 and 158 µm, respectively. The chitosan hydrogel matrix itself had some particulates on the surface which is considered due to sodium salts from neutralising agent. In CS/HA hydrogels the surface of chitosan matrix was covered by the spherical hydroxyapatite particles which are considered to provide active surface properties (Figure 7b) 

The hydrogel composites with heparin has also porous morphology with a folding structure on top surface, as can be seen for SIII sample at Figure 8a. The semi-crystalline chitosan network involving the prismatic shaped crystals can be attributed to sodium salts result from neutralizing agent. The thinner round like hydroxyapatite particles were detected as agglomerates at different regions of hydrogels. The smoother surface feature covering all rough particles might be the indicator of coating effect of heparin on the surface (Figure 8b–d). 

### 2.5. Chemical Analyses 

#### 2.5.1. Fourier-Transform Raman (FT-Raman) Spectroscopy

The FT-Raman spectra of all gel compositions are given at Figure 9. The peaks at the region of 3100–3400 were assigned as hydroxyl groups while a sharpest peak at 2932 and its small shoulder at 2748 cm^−1^ indicate C–H stretching peaks in all samples. The peaks at 960 and 1035 cm^−1^ showed the presence of (PO_4_)^−3^ bonds in hydroxyapatite [64]. The S=O bonds at 1260 cm^−1^ become the peak has become sharper and more intense showing the characteristic of heparin in compare to control samples [63]. The intensity of the peak at 446 cm^−1^ has increased in higher heparin concentrations due to O=S=O bonds in heparin [65].

#### 2.5.2. Attenuated Total Reflectance (ATR) Spectroscopy 

Figure 10 presents the overlay of ATR spectral data of all gel compositions. The peaks between 3000–3400 cm^−1^ were assigned as OH and NH stretching [66,67]. C–H stretching of chitosan and heparin were observed at ~2930 and 1150 cm^−1^, which decrease when the heparin concentration increases. The peaks near to 1020 cm^−1^ correspond to ring stretching and in plane bending of C–H in CS, asymmetric stretching of PO_4_^−3^ bond [66,68].

The pyranoid ring stretching of chitosan was seen at 898 cm^−1^ at CS (CI) sample while this value was slightly decreased (893 cm^−1^) gradually when the Hep amount increased [67]. In addition, pyranoid ring stretching of C–H and symmetric COO^−^ vibrations occurred at 927 cm^−1^ [66,69]. Amide II or N–H bending and C–H deformation occurred in most composition at 1544 and 1406 cm^−1^, respectively [66,67]. Amide III or C–N stretching in CS and was at 1338 cm^−1^ [70]. Amide I groups have taken place at 1636 and 1687 cm^−1^ [66,67].

### 2.6. Ex-Ovo Chick Chorioallantoic Membrane (CAM) Assay 

Angiogenesis evaluation of hydrogels with and without heparin was carried out with an ex-ovo (shell-less) CAM assay on chick embryos (see Section 4.4.7). After implementing hydrogel samples on CAM at day 7 of the incubation, images of newly formed blood vessels attached to samples were taken at day-10 and day-14. The obtained results have revealed pro-angiogenic response of hydrogels leading micro vascularity at adjacent tissues to the samples. The total survival rate of CAM experiments and the survival rate after sample implantation were between 45–50% and 70–80%, respectively. The Figure 11 shows a comparison of new blood vessel occurrence with the effect of different concentration of heparin with control (CII). 

#### Vascular Index Counting 

The newly formed blood vessels at the images acquired from CAM (day-14) were counted by vascular index method by Barnhill and Ryan (1983) [71]. Vascular index counted for six samples from each sample group were plotted in GraphPad Prism (Version 7.0, San Diego, CA, USA) and unpaired two-tailed t-test applied.

The results have shown that SI hydrogels with minimum heparin concentration (0.12 mg/mL in final sol) has shown maximum pro-angiogenic response in the CAM (Figure 12). However, micro vessel formation in control samples (CII:CS/HA) without heparin has also comparable with SI sample. In addition, the highest concentration of Hep (SIII) has led to less vascularity compared to control (CII). 

### 2.7. Histology Analyses of CAM Tissue-Samples

Histology sections of hydrogel-tissue samples preserved after CAM assay were acquired after fixing, sectioning and Hematoxylin and Eosin (H&E) staining. Figure 13 shows images of horizontally sectioned slides obtained from a CII hydrogel-tissue specimen. Images showed that there is a connective tissue formation in the edge of circular hydrogel specimen. Blood vessels filled with blood cells grooving toward to the samples can be seen clearly. These results indicate also well integration of hydrogels with surrounding tissues since hydrogels found in the connective tissue regions which are adjacent newly growing micro vessels. In addition, endothelial cells and other blood cells were located in hydrogels.

## 3. Discussion

Injectable homogeneous chitosan-based hydrogels solutions were obtained, and gelation was occurred at the vicinity of body temperature. Qualitative injectability rating evaluated with a range of needle coupled with 10 mL syringe showed all present compositions had good drop flow from 21 Gauge and thicker sized needles. Therefore, viscosity of solutions is found suitable for injectability in bone regeneration applications (orthopaedic procedures and dental applications are 10–16 G and 16–25 G, respectively) [72]. Furthermore, the quantitative measurements have shown that the maximum force applied during the tests with 19 G needle was 17.191 ± 2.296 N which is less than the maximum manual force for injection is considered as 30 N [73]. 

In final mixture solutions, the maximum solution pH after neutralization with NaHCO_3_ was less than 6.4. To avoid heterogenicity in the solution, pH was kept at the vicinity of 6.2 but the gelation started in 5 to 10 min and this led to increase at the pH up to 7 which is in the range of physiological pH level [33]. As has been reported by other authors [41] although pH could be increased above 6.5 by increasing sodium bi-carbonate amount in chitosan solution, the phase separation occurred during the storage of solution following several days. Although the utilization of β-GP lead higher pH values in thermosensitive chitosan-based hydrogels, cell cytotoxicity problem has been reported [36]. Therefore, recently researchers have harnessed sodium bicarbonate accompanying β-GP which has led to increase at cell viability [42]. Other researchers have reported thermosensitive chitosan hydrogels with co-synthesised hydroxyapatite which were neutralised by only sodium bicarbonate. The results indicate the beneficiary impact of hydroxyapatite at surging pH up to 6.96 [40]. However, as reported in these formulations, the incipient gelation point could not be determined by the cross over point of elastic and viscous modulus since at the beginning of the rheology measurements elastic modulus was higher than the viscous modulus. Although this result shows desirable pH values in solutions and mechanical strengthen hydrogels, having viscous solutions can cause limitations in injectability. Therefore, in CS/HA/Hep hydrogels, increase at the pH can be provided by involving higher amount of hydroxyapatite. However, injectability and gelation features need to be preserved in the meantime.

Rheology tests have revealed that the sol–gel transition of all hydrogel compositions showed thermo-irreversible gelation feature. These thermosetting behaviours were distinguished from chitosan hydrogels neutralized by β-GP which presents thermo-reversible sol–gel behaviour at pH values between 6.5–6.9 despite their irreversibility at higher pH [33]. Thermosetting hydrogels can be considered as an advantageous that once the homogeneous sol easily injected into tissue defects, hydrogels form and they strengthened mechanically with time. This feature can be suitable for sustainable drug delivery applications. The rheology studies have also showed that the gelation time and temperature can be modulated by hydroxyapatite and heparin inclusion since both compounds have contributed to decrease at the gelation temperature. The highest concentration of heparin (0.60 mg/mL in final solution) has led to almost 30-fold increase at the elastic modulus of the hydrogels. This mechanical enhancement can be attributed to strong reversible ionic interaction chitosan and heparin which is due to the oppositely charged polyelectrolyte complex phenomena [74].

The heparin inclusion method inside hydrogels is also very important in terms of stability of heparin and blood compatibility as well as homogeneity of sol in CS/Hep polycomplex. Polymer mixing method was found suitable by authors [75] since it provides immobilization of heparin leading sustainable delivery of heparin as enhancing blood biocompatibility in CS/Hep composites. In addition, they stated that the formation of a homogeneous solution from CS/HA polycomplex depends on the molecular weight of chitosan, the mixing order of compounds and solid or liquid status addition, and speed of mixing. In our hydrogel system (CS/HA/Hep), the synthesis method was optimised to get homogeneous solutions. While use of heparin in powder state lead to huge precipitation problem, the gradual addition of heparin as water solution with constant stirring ensured to acquire homogeneous solutions. In addition, as has been reported [75], the use of medium molecular weight (Mw) of CS in CS/Hep polycomplexes result with homogeneous solution whereas low and high Mw CS lead to precipitates in solutions. Our results have also confirmed this positive effect of medium Mw CS providing homogeneity compared to other molecular weights of CS. 

Hydrogel composites in freeze dried state has revealed highly porous and interconnected pore morphology varying between 19.5 and 158 µm while average pore diameter is 73.4 µm. Although freeze drying process can cause alteration in hydrogel structures leading less porosity than that of swollen hydrogels [56], the representative porosity of hydrogels in the freeze-dried form are also comparable with the pore sizes required to maintain cell activities and diffusion in bone tissue engineering (75–100 µm) [76]. 

Angiogenesis evaluation by ex-ovo CAM assay have shown that composite hydrogel samples with or without heparin had exhibited pro-angiogenic response. Histological analyses have also shown the evidence of the blood vessel generation in a spoke-wheel like pattern in the edges of hydrogels assigned as pro-angiogenic response [77]. Although the vascular index of the minimum heparin loaded samples (0.12 mg/mL) was slightly higher than sole CS/HA hydrogels, the difference was not significant. The CS/HA samples were alone also exhibited pro-angiogenic response in CAM. In addition, the vascular index has decreased with the highest concentration of Hep. In this study, Hep concentration range was higher as compared to the previous studies that Hep has involved through direct mixing into solutions [63]. Additionally, there is very limited studies investigating different concentrations of Hep on angiogenesis. In one study for cardiovascular biomaterial application, it has been reported that high Hep concentrations have decreased the vascular cell proliferation while low heparin amount triggered the endothelial progenitor cell production and endothelial cell proliferation and but reduced the smooth muscle cell proliferation [78]. In another study, researchers have reported hemocompatible Hep-CS coatings on the decellularized bone scaffolds leading the blood diffusion, endothelial generation and bone vascularization [79]. Although the current lowest concentration of Hep in CS/HA hydrogel system was found the best concentration among all, the less concentrations of Hep may lead better pro-angiogenic response. The lower Hep concentrations in compositions are under further investigations in our lab. Overall, in terms of injectability and gelation features and pro-angiogenic response of most thermosensitive hydrogel compositions, CS/HA/Hep bioactive natured injectable hydrogels could have potential applications as minimally-invasive biomaterials to promote vascularized bone tissue regeneration. 

## 4. Materials and Methods 

### 4.1. Materials

For preparation of thermosensitive hydrogels, the raw commercial hydroxyapatite (HA) (CAPTAL^®^ ‘R’ grade, Batch No: P218R) was supplied from Plazma Biotal (Buxton, UK). Chitosan (medium molecular weight, 100–300 kDa) and glacial acetic acid (Aca) (ACROS Organics^TM^, Thermo Fisher Scientific, Geel, Belgium) and sodium bi-carbonate (NaHCO_3_) (Fluka^®^, Sigma Aldrich, USA) were purchased, and injectable grade sodium bovine heparin was kindly gifted by Extrasul Ext. An. Veg. LTDA (Sao Paulo, Brazil).

### 4.2. Synthesis of Thermosensitive Injectable CS Hydrogels 

Prior to the reactions, reagent solutions were filter sterilized by using 0.22 µm polyethersulfone (PES) membrane filter. Synthesis of hydrogels were carried out in aseptic conditions. In control solution sample (CI), initial CS concentration of 5% and 6% (*w*/*v*) (which account for 2.44% and 2.93% (*w*/*v*), respectively, in the final sol) were found optimal in terms of injectability and gelation features. Therefore, all composite hydrogels (CS/HA/Hep) were prepared with the final chitosan concentration of 2.44% (*w*/*v*).

Initial experiments covered the optimisation of CS concentration in terms of the final sol injectability after pH neutralisation and gelation capability of final sol. Therefore, different concentrations of CS solutions were prepared in acetic acid including 4%, 5%, 6% and 8% (*w*/*v*). Among these concentrations, chitosan initial solution of 5% (*w*/*v*) was chosen as optimal concentration which is approximately 2.4% (*w*/*v*) in final solution after neutralisation by sodium bicarbonate. The further synthesis of composite hydrogels was carried out by using this concentration.

For preparation CS hydrogel solutions, 1 g of CS was dissolved in 20 mL, 0.5 M acetic acid by maintaining constant stirring. Before pH neutralisation with NaHCO_3_, chitosan solution was chilled in a fridge at 4 °C for 15 min. Then, the stock NaHCO_3_ solution (0.48 M) 4 °C was dropwise added into the chilled chitosan solution in a flask placed into an ice bath at 4 °C. Homogeneous solutions were obtained after pH adjustment up to (6.2–6.3) by ensuring constant stirring.

### 4.3. Synthesis of Thermosensitive Injectable Composite Hydrogels (CS/HA/Hep) 

For CS/HA hydrogel preparation, medical grade HA powders were mixed into CS (5% *w*/*v*) solution (HA/CS:1/10), and this dispersion was kept stirring overnight to obtain a homogenous mixture. Then, the solution pH was neutralised as described in Section 4.2.

CS/HA/Hep hydrogels with three different concentrations: SI, SII and SIII with the concentration of 0.45%, 1.35%, and 2.22% (*w*/*w*) Hep, respectively, were prepared (See Table 1). The sterile Hep solution in respective amounts was prepared by dissolving in deionized water. Then, Hep solution was gradually added into CS/HA solutions. Subsequently, pH of the CS/HA/Hep solution was adjusted via gradual addition of NaHCO_3_ solution in an ice bath. In Figure 14, the synthesis of injectable composite (CS/HA/Hep) and pH and thermosensitive gelation mechanism are presented. 

Homogeneous hydrogel solutions were placed into universal tubes with closed lid and stored in a fridge at 4 °C. Hydrogels were formed in glass test tubes or glass dishes with lids upon incubation of solutions at 37 °C. 

### 4.4. Characterisation of Hydrogels

Hydrogel solutions were freeze-dried at −20 °C in a freeze drier (SciQuip Epsilon 1-4 LSC, Shropshire, UK) prior to the morphological and chemical characterisations.

#### 4.4.1. pH Measurements

The pH of reagent solutions and their mixtures were measured by using a pH-meter (Mettler-Toledo GmbH, Analytical, FiveEasy Plus™, Greifensee, Switzerland).

#### 4.4.2. Injectability Measurements of Hydrogels

Injectability of thermosensitive solutions were tested in two different ways. The first method was the qualitative evaluation of the flow rates of each solution from a range of needle sizes. The second method was quantitative measurements of diverse hydrogel solutions in terms of force applied for injection. 

##### Qualitative Injectability Measurements 

Injection capacity of all solutions were tested immediately after the synthesis, by using 10 mL disposable syringes (BD, Luer-Lok^™^, (Becton, Dickinson U.K. Limited, Wokingham, UK) with a series of needle sizes; 18G ×1 1/2” (1.2 × 40 mm), 19G ×1 1/2” (1 × 40 mm), 20G ×1 1/2” (0.8 × 40 mm), 21G ×1 1/2” (0.8 × 40 mm), 22G ×1 1/2” (0.7 × 40 mm), 23G ×1” (0.6 × 25 mm), and 25 G ×5/8” (0.5 × 16 mm). The injection observations for each solution were rated comparatively. 

##### Quantitative Injectability Measurements 

Quantitative measurement of injectability was carried out by a Universal testing Machine (Instron^®^ 3345, Norwood, MA, USA) in compressive mode adapted as described in the literature [80,81,82]. A 10 mL syringe (BD, Luer-Lok^™^) was filled with 5 mL hydrogel solution and fixed into a syringe rig which was placed between the compression plates of the machine. Experiments were carried out by using the syringe coupled with a needle (19 G ×1 1/2” 1 × 40 mm) and without needle. After the contact of the upper compression plate onto the plunger, a 30 mm displacement of the plunger at the speed of 0.5 mm/s was applied to expel 5 mL of solution through the syringe. The force (kgf) versus displacement (mm) measurements were obtained in Bluehill^®^ Universal Software (Version 4.06, Norwood, MA, USA).

Experiments were repeated three times for each solution. The force unit was converted into SI unit, Newton counted in the Excel. The following forces were counted as mean values of three experiments with standard deviation; initial glide force (IGF): the force required to initiate syringe plunger motion, dynamic glide force (DGF): the force required to maintain the plunger movement and the maximum force (F_max_) during plunger displacement. 

##### Test Tube Invert Method 

Test-tube invert method was used to identify initial gelation time of the mixtures. In this method, 2 mL of the solution was taken into a closed glass tube (with push polythene plug cap, 75 mm × Ø:18 mm) placed in a beaker filled with silicon oil bath on hot plate at 37 °C. The tube was tilted horizontally every minute, and then the time when the solution did not have any flow was recorded as incipient gelation time (t_i.g._). 

#### 4.4.3. Rheology Measurements

Rheological analyses of hydrogels were performed using a Rheometer (TA Instruments^®^, TA-AR 2000, Newcastle, DE, USA) with a cone geometry (Ø = 40 mm, 2°). Frequency sweep tests were carried in a viscoelastic region at a constant strain of 1%, the frequency range of 0.1–10 Hz at certain temperatures; 25 °C, 15 °C, and 37 °C. The storage (G′) and loss modulus (G″) changes were recorded at 1 Hz constant frequency in oscillatory measurements during heating and cooling between 15–37 °C temperatures. In addition, temperature was held at 37 °C for 30 min to observe the alteration of the viscoelastic behaviour of hydrogels with time. Incipient gelation point of solutions was determined via the intersection point of G′ and G″ (tan δ:G″/G′ = 1) when the material start to present more viscous behaviour rather than elastic (sol–gel transition) [83]. 

#### 4.4.4. Scanning Electron Microscopy (SEM)

Freeze dried hydrogels were gold coated (10 nm) and morphological images were obtained by using an SEM device (FEI Quanta^TM^ 650 3D FEG/FIB ESEM^TM^, Hillsboro, OR, USA). The image analyses were performed by image-J^®^ (Version 1.52 k, National Institutes of Health, Bethesda, MD, USA) software by manual measuring a series of pores and counting the mean, minimum and maximum lengths of individual and interconnected pores.

#### 4.4.5. Fourier-Transform-Raman (FT-Raman) Spectroscopy

FT-Raman analyses were carried out by using Thermo Nicolet^TM^ iS50 FTIR spectrophotometer (Thermo Fisher Scientific Inc, Madison, WI, USA) in conjunction with FT-Raman compartment. The measurements were carried by using Detector InGaAs and Beam splitter CAF_2_. In these experiments following experimental parameters were used: Aperture: 200; resolution: 16 cm^−1^; scan number: 64; and the wavelength range: 4000–400 cm^−1^. Spectral data were obtained via a nine-points square area mapping, and the resultant spectra was obtained with the average of all spectra in Thermo Nicolet OMNIC^TM^ version 9 software.

#### 4.4.6. Fourier Transform Infrared (FTIR) Spectroscopy

FTIR spectra of hydrogel samples were obtained using Thermo Nicolet^TM^ iS50 FTIR spectrophotometer (Thermo Fisher Scientific Inc, Madison, WI, USA) in conjunction with Attenuated Total Reflectance (ATR) sampling accessory to allow the analysis of neat samples without any sample preparation. Experiments were performed with DTGS ATR detector and KBr beam splitter in the mid-infrared region (4000–400 cm^−1^) at 16 cm^−1^ resolution accumulating 128 scans. Thermo Nicolet OMNIC^TM^ software was employed for data acquisition. 

#### 4.4.7. Ex-Ovo Chick Chorioallantoic Membrane (CAM) Assay 

Chick Chorioallantoic Membrane (CAM) assay is very useful technique to investigate angiogenic response of biomaterials and biocompatibility [71]. A standardised ex-ovo (shell-less).

CAM assay protocol was followed as described by authors [84]. CAM assay was used to explore the angiogenic potential of CS/HA/Hep hydrogels. A representative ex-ovo CAM technique and related analyses are illustrated in Figure 15. To elaborate, first fertilized brown eggs were purchased from (Henry Stewart & Co.Ltd, Louth, UK) on the lay day and were incubated horizontally for 3 days in rotational incubators (RCOM King SURO, P&T Poultry, Powys, Wales) with 60% humidity at 37.5 °C. At day 3, by using ex-ovo technique, eggshells were removed and placed embryo into sterilised weighing boats including 2 mL of 1% penicillin/streptomycin solution (Sigma-Aldrich^®^, Dorset, UK) prepared in Phosphate Buffer Saline (PBS-Dulbecco A, Thermo Scientific™, Oxoid™, Basingstoke, UK). Then, chick embryos were kept in static incubators at 37.5 °C. 

At day 7, implantation was carried out on CAM. Prior to implantation, hydrogel solutions in bijous were sterilized under the UV in an ice bath to prevent gelation in the room temperature. By using a micro-syringe, 100 µL of hydrogel solution was dropped on CAM, between two arteries and at the midway of the chick embryo’s formation and at the shiny CAM border. Subsequently, thermosensitive hydrogels were formed on chick CAM membranes in-vivo upon incubation at 37.5 °C forming gel layer from surface in minutes. Then, newly formed growing blood vessels at the vicinity of hydrogels were photographed with a scale, at day 10 and 14 of incubation. At day 14, embryos were sacrificed, and samples were retrieved and fixed with formaldehyde for further histology studies. The wastes were disposed accordingly Laboratory regulations for animal waste.

The angiogenesis was quantified by using Image-J^®^ (Version 1.52 k, National Institutes of Health, Bethesda, MD, USA) Software through the method defined as ‘vascular index’ as described by Barnhil and Ryan [71]. Angiogenesis was evaluated for all hydrogel specimens with and without heparin as control.

#### 4.4.8. Statistical Analysis

Comparative results of blood vessel counts were plotted in Graph Prizm for six samples per group and analysed by GraphPad Prism (Version 7.0, San Diego, CA, USA) and unpaired two-tailed t-test applied.

#### 4.4.9. Histology 

The tissue-hydrogel specimens retrieved after CAM assay were fixed by using 3.7% formaldehyde (Sigma-Aldrich^®^, Dorset, UK) prior to histology studies. Then, tissue samples were moulded by using a tissue freezing media (Leica, Germany) and subsequently were snap frozen by using liquid nitrogen. The frozen tissues were sectioned in 10 µm thickness by using a cryostat (Leica Biosystems, CM1860 UV, Nussloch, Germany). Then, the standard Haematoxylin Eosin staining protocol with supplied reagents (Sigma-Aldrich^®^, Dorset, UK) was performed. The stained specimen slides were imaged by using optical microscope (Motic, Barcelona, Spain) to identify tissue-sample interactions and vascular tissue features.

## 5. Conclusions

The functionalised novel bioactive injectable CS/HA/Hep hydrogels were prepared via sol–gel technique. Homogeneous injectable solutions were acquired by indirect gradual mixing of Hep as solution, into hydroxyapatite dispersed medium Mw CS solution. Injectability capacity of formulations accords with manual injection for dental and orthopaedic applications, and pH formation of thermosensitive hydrogels start in 5 to 10 min and set at 37 °C. Interconnected and porous morphology of freeze-dried hydrogels ranging pores from 19.5 µm to 158 µm with average pore size 73.4 µm provide maintenance of cell activities and diffusion, as well as angiogenesis for bone regeneration. 

Polymer blending method is beneficial to stabilize Hep in CS/Hep polycomplex leading better mechanical strengthening due to affinity bonding, and sustainable release of Hep or delivery of other targeted active agents. The highest concentration of Hep (0.60 mg/mL) in hydrogels led to increase in elastic modulus almost 30-fold whereas initial gelation temperature has decreased up to 21 °C which can lead quicker gelation compromising injectability. In addition, the angiogenesis via these samples was less than the sole CS/HA hydrogels. 

The vascular index obtained via minimum heparin loaded samples (0.12 mg/mL) was the highest, however sole CS/HA hydrogels also showed the comparable results. This might be associated with chosen a high concentration range of Hep in comparison to those of in similar studies. As previously addressed, the low concentration of Hep inducing production of progenitor cells and endothelial cell proliferation while high concentrations lead decrease in the vascular cells had been reported. In conclusion, CS/H/Hep hydrogels in current minimal Hep concentration induces angiogenesis which might be enhanced via lower concentrations.

To conclude, minimally invasive in-situ forming bioactive CS/H/Hep hydrogels versatilely functioned to induce angiogenesis, which could be employed for rapid bone regeneration and drug delivery applications.

## Figures and Tables

**Figure 1 ijms-21-01633-f001:**
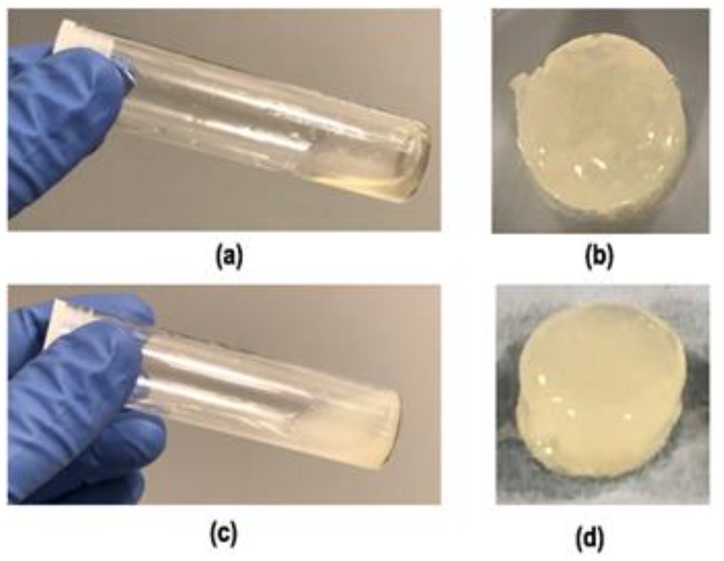
Images of the final homogeneous solutions (**a**) transparent CI (CS, 5% (*w*/*v*) initial concentration) and (**c**) composite sol, SI: CS-0.1HA-0.005Hep (solutions were obtained after neutralising with NaHCO_3_ and 2 mL of the solution was taken into a glass-tube (75 mm × 18 mm) with three level push polythene plug cap) and hydrogels of (**b**) CI and (**d**) SI acquired after heat treatment in one level open plug cap of the tube at 37 °C-oven for 48 h.

**Figure 2 ijms-21-01633-f002:**
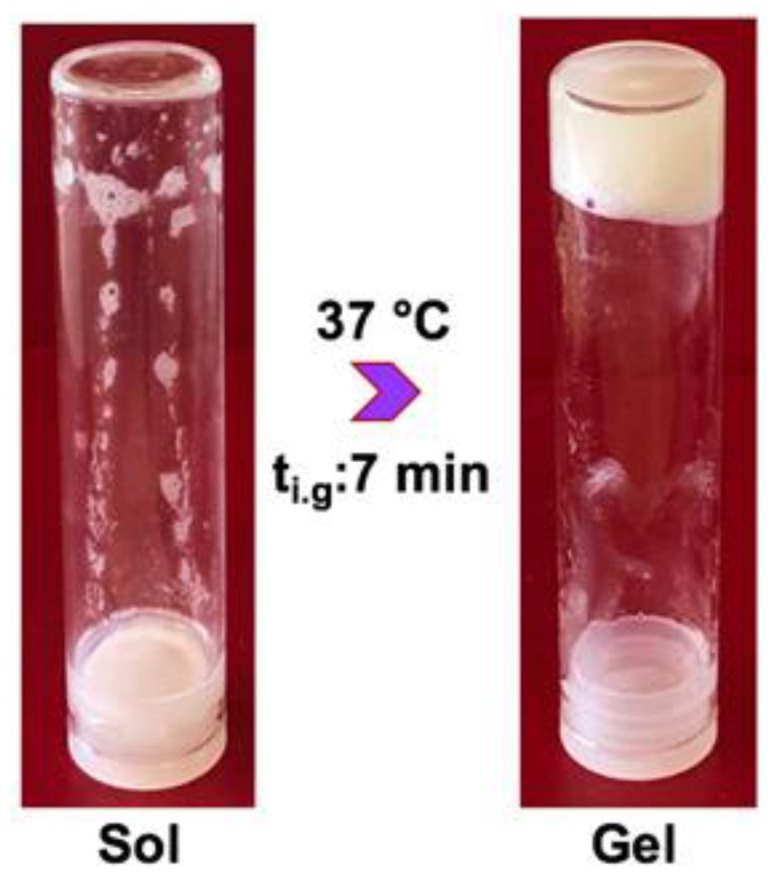
A representative sol–gel transition image of a composite SI hydrogel solution (2 mL) in a closed glass test tube (75 mm × 18 mm) placed into a silicone oil bath at 37 °C. t_i.g._*:* incipient gelation point was determined with the test tube invert method as 7 min.

**Figure 3 ijms-21-01633-f003:**
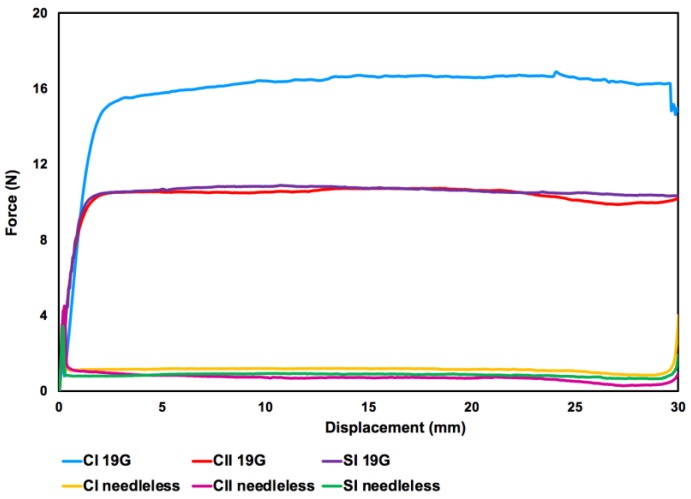
A representative graph of the injectability tests of hydrogel solutions via force versus displacement measurements. CI/II—define; 19G—define; SI—define.

**Figure 4 ijms-21-01633-f004:**
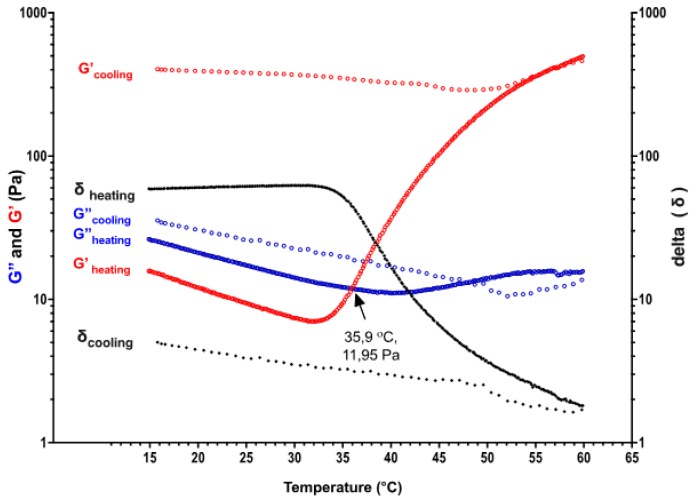
The gradient of viscous and elastic modulus of thermosensitive CS sol (initial concentration of CS solution 6% (*w*/*v*) neutralised by NaHCO_3_) with temperature sweep between 15–60 °C during heating and cooling, respectively.

**Figure 5 ijms-21-01633-f005:**
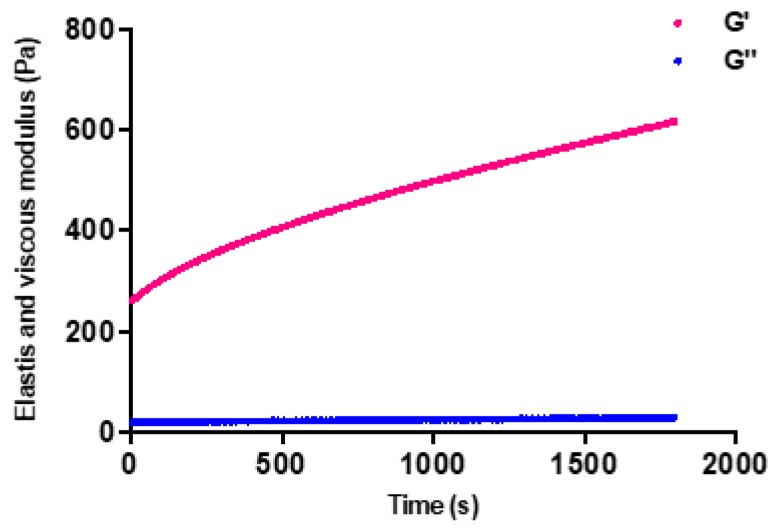
Elastic and viscous modulus changes versus time with a constant temperature at 37 °C for of CI:CS sol.

**Figure 6 ijms-21-01633-f006:**
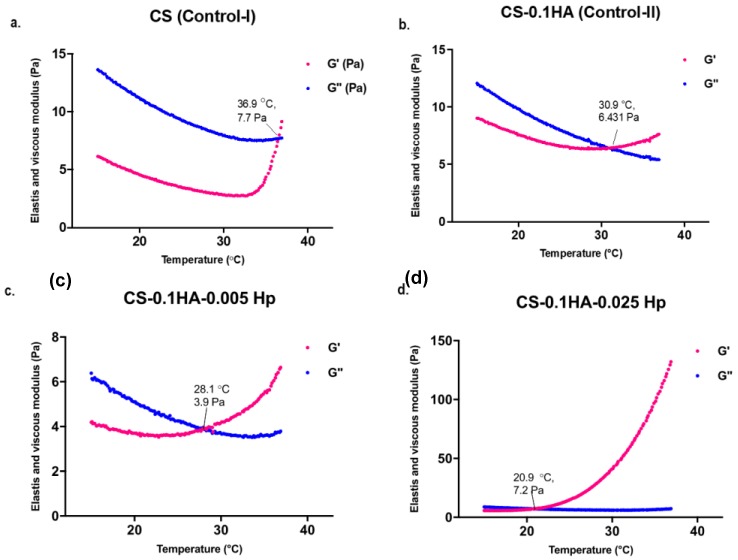
Elastic and viscous modulus changes with the increasing temperature between 15 and 37 °C in following hydrogel compositions: (**a**) CI: CS; (**b**) CII: CS-0.1HA; (**c**) SI: CS-0.1HA-0.005Hep; (**d**) CS-0.1HA-0.025Hep.

**Figure 7 ijms-21-01633-f007:**
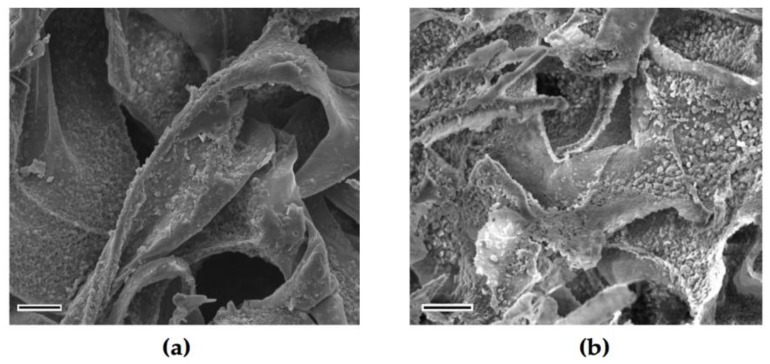
SEM micrographs of lyophilized hydrogels: (**a**) CI:CS; (**b**) CII: CS.0.1HA at 5000 magnification (Scale bars indicate 10 µm).

**Figure 8 ijms-21-01633-f008:**
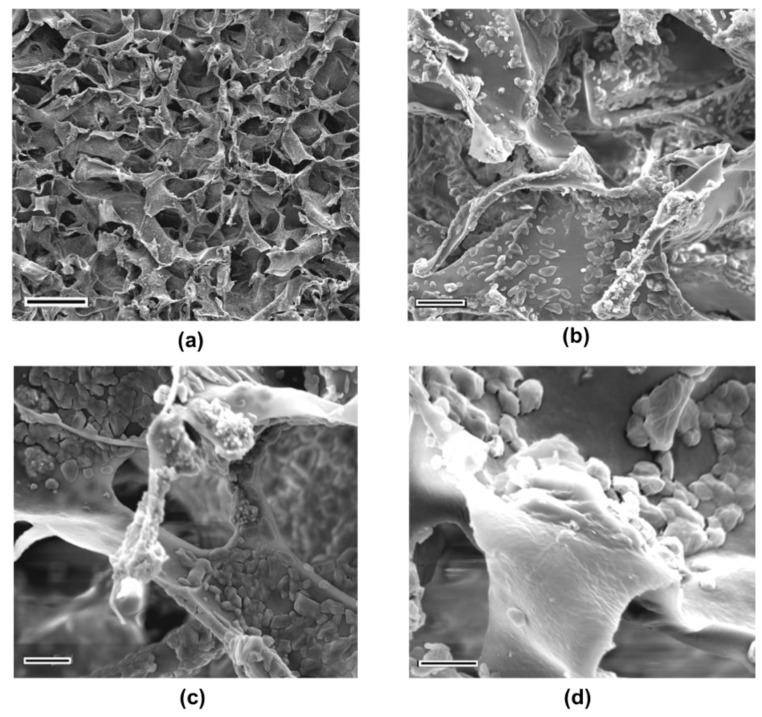
The SEM images of SIII:CS-0.1HA-0.25Hep at different magnifications; (**a**) 500 (Scale bar is 100 µm); (**b**) 5000 (Scale bar is 10 µm); (**c**) 10,000 (Scale bar is 5 µm); (**d**) 20,000 (Scale bar is 3 µm).

**Figure 9 ijms-21-01633-f009:**
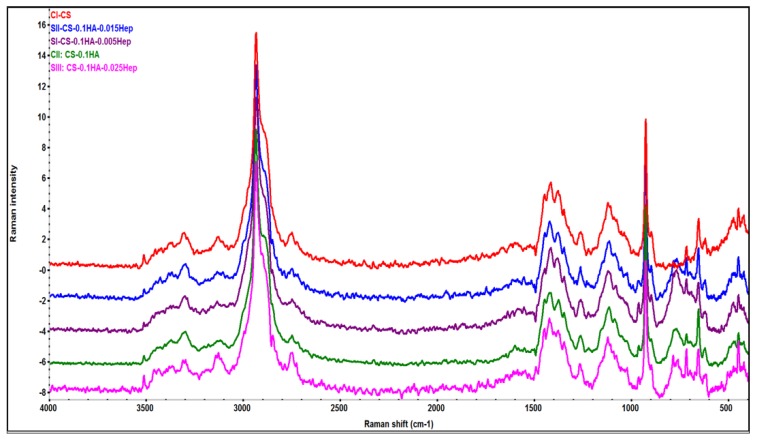
A comparison of FT-Raman spectra of all hydrogel samples obtained by mean of nine spectra from a square area mapping and baseline corrected and offset data in OMNIC^TM^ software.

**Figure 10 ijms-21-01633-f010:**
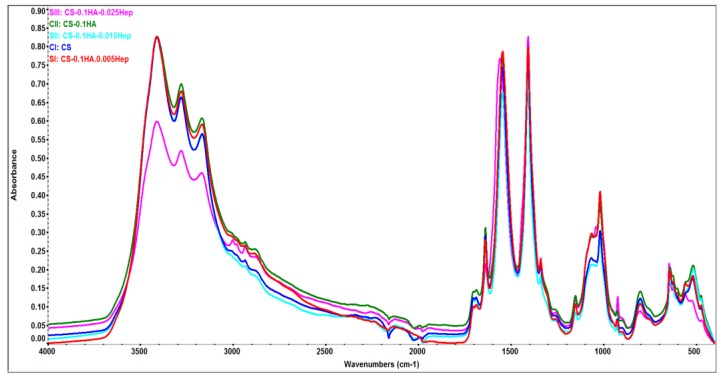
Surface chemical analyses of all hydrogel samples by FTIR-ATR spectroscopy. Data has been baseline corrected in OMNIC ^TM^ software.

**Figure 11 ijms-21-01633-f011:**
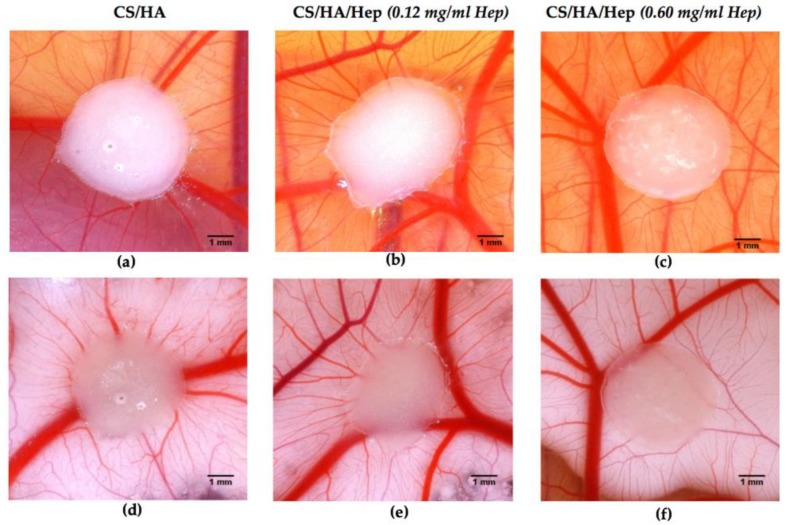
Comparative chorioallantoic membrane (CAM) images taken at day-14 of embryo incubation to evaluate angiogenic response of hydrogel samples: (**a**) CII: CS.0.1HA; (**b**) SI: CS.0.1HA.0.005Hep (0.12 mg/mL Hep); (**c**) SIII: CS.0.1HA.0.025Hep (0.60 mg/mL Hep), and images after white cream injection underneath CAM to make blood for contrast: (**d**) CII; (**e**) SI; (**f**) SIII.

**Figure 12 ijms-21-01633-f012:**
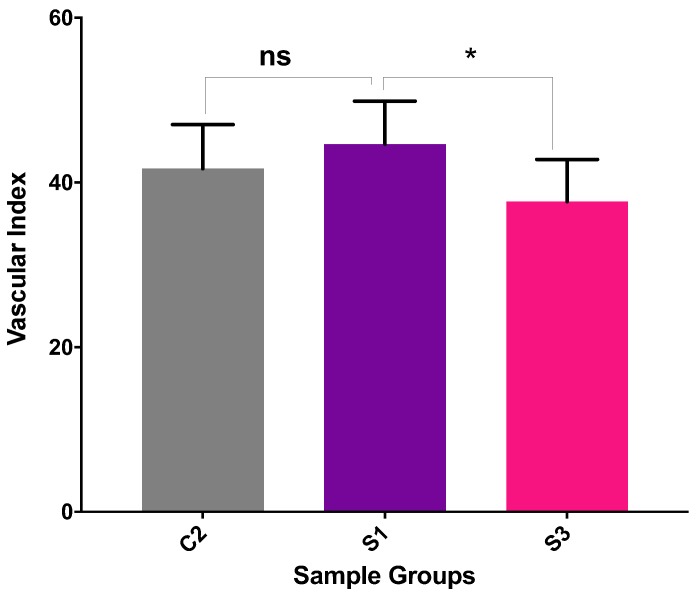
Quantification of angiogenesis on the images taken in the sacrificing day of embryos (day-14) in the following samples: C2: CS.0.1HA; S1: CS.0.1HA.0.005Hep (0.12 mg/mL Hep); S3: CS.0.1HA.0.025Hep (0.60 mg/mL Hep) (ns and * represent the *p* values which are equal to 0.3482 and 0.0408, respectively).

**Figure 13 ijms-21-01633-f013:**
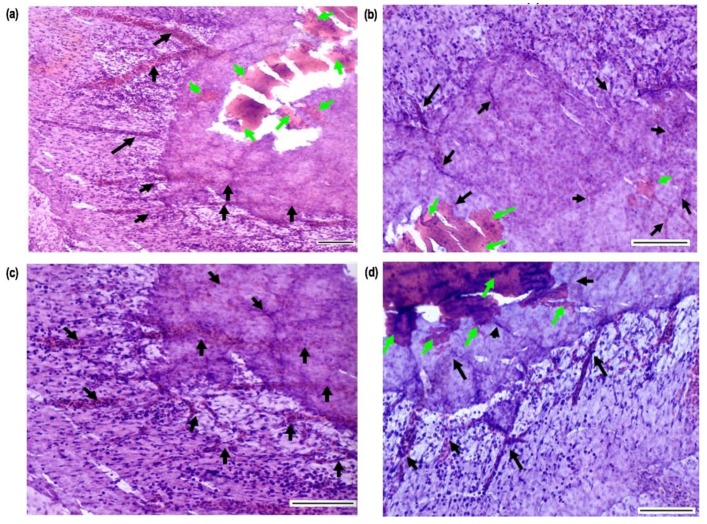
Histology images of CAM tissue-hydrogel slides obtained after horizontal sectioning and H&E staining: Images represents the connective tissue surrounding hydrogels with tiny blood vessels grooving into them (Images were taken at a magnification of (**a**) ×10 and (**b**–**d**) at ×20, and all scale bars equal to 10 µm. The green and black arrows represent hydrogels, and blood vessels, respectively).

**Figure 14 ijms-21-01633-f014:**
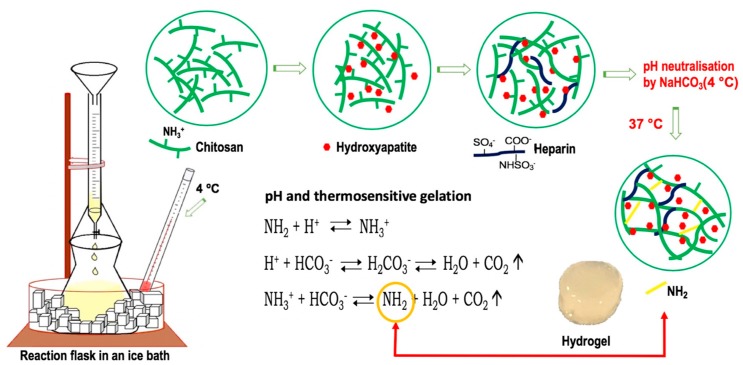
Schematic illustration of synthesis steps of pH dependant ionically crosslinked mechanism of bioactive composite based thermosensitive hydrogels (CS/HA/Hep).

**Figure 15 ijms-21-01633-f015:**
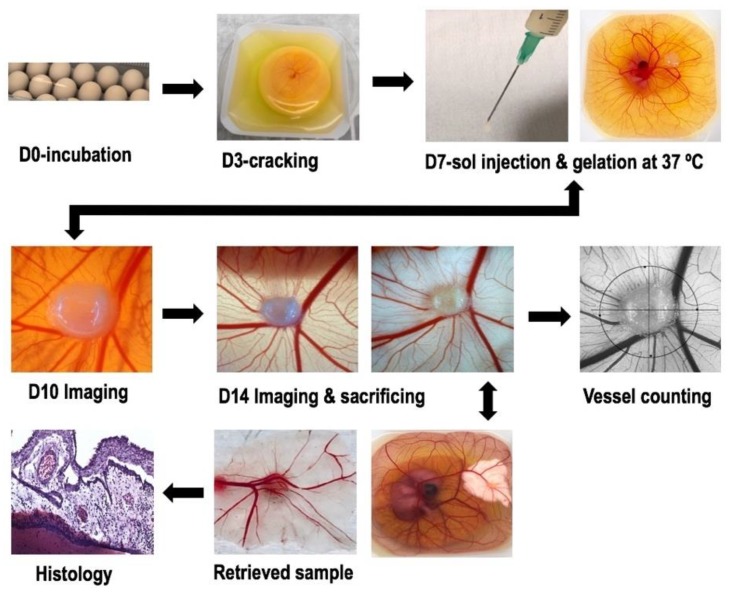
The process diagram for ex-ovo CAM assay technique to investigate pro-angiogenic potential of pH and thermosensitive injectable CS/HA/Hep hydrogels, vascular index counting in Image-J^®^ (Version 1.52 k, National Institutes of Health, Bethesda, MD, USA) Software, and example histology sectioned sample.

**Table 1 ijms-21-01633-t001:** Different hydrogel compositions according to their weight ratio, pH values of solutions and hydrogels and incipient gelation time in test tubes at 37 °C.

Sample Names and Codes	Chemical Compositions (*w*/*w*, %)			Incipient Gelation Time (t_ig_), min	pH of Final Sol	pH after Gelation
	CS	HA	Hep			
CI: CS Gel from pure CS	100.00	0.00	0.00	8	6.20	6.50
CII: CS.01HA (CS+HA; HA/CS=1/10)	90.91	9.09	0.00	7	6.24	6.51
SI: CS-0.1HA-0.005Hep (CS+HA+Hep; 0.12 mg/mL Hep)	90.50	9.05	0.45	5	6.21	6.53
SII: CS-0.1HA-0.015Hep (CS+HA+Hep; 0.36 mg/mL Hep)	89.69	8.97	1.35	10	6.22	6.72
SIII: CS-0.1HA-0.025Hep (CS+HA+Hep; 0.60 mg/mL Hep)	88.89	8.89	2.22	7	6.24	7.01

CI—Chitosan gel; CS—Chitosan; HA—hydroxyapatite; SI = Chitosan; Hep—Heparin; Sol—definesolution.

**Table 2 ijms-21-01633-t002:** Injectability rating of different hydrogel compositions ^1^.

Sample Names and Codes:	Needle Sizes (Gauge)						
	18	19	20	21	22	23	25
CI: CS	******	******	******	*****	****	***	**
CII: CS-0.1HA	******	******	******	*****	****	***	**
SI: CS-0.1HA-0.005Hep	******	******	******	*****	****	***	**
SII: CS-0.1HA-0.015 Hep	******	******	******	*****	****	***	**
SII: CS-0.1HA-0.025 Hep	******	******	******	*****	***	***	**

^1^ ******—a perfect flow; *****—a good drop flow; ****—a drop flow with slight pressure; ***—a drop flow with high pressure; **—a drop flow with very high pressure; *—no flow.

**Table 3 ijms-21-01633-t003:** Injectability force measurements of hydrogel solutions by 10 mL syringe (BD, tip inner and outer diameter of 2 and 4 mm, respectively) with and without a needle (19G × 1 1/2” 1 × 40mm). Initial glide force (IGF), dynamic glide force (DGF) and maximum force (F_max_) values were counted as mean of three experiments (*n* = 3) with standard deviations in Newton (N) unit.

Syringe Type	Sample	IGF (N)	DGF (N)	F_max_ (N)
Needleless	CI	4.478 ± 0.273	1.217 ± 0.050	4.478 ± 0.273
	CII	3.219 ± 0.331	0.679 ± 0.033	3.219 ± 0.332
	SI	3.138 ± 0.054	1.36 ± 0.050	3.138 ± 0.054
19G needle	CI	2.697 ± 2.296	16.697 ± 0.324	17.191 ± 2.296
	CII	4.520 ± 1.125	10.682 ± 0.265	10.996 ± 1.125
	SI	3.127 ± 1.263	11.460 ± 0.172	11.892 ± 1.263

IGF— initial glide force; DGF— dynamic glide force; F_max_— maximum force.

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
