# Peer review of "In-Situ Forming pH and Thermosensitive Injectable Hydrogels to Stimulate Angiogenesis: Potential Candidates for Fast Bone Regeneration Applications"

_ijms, 2020, doi:10.3390/ijms21051633_

Round 1
Reviewer 1 Report
Dear authors,
A more comprehensive graphical abstract, or an additional figure which explains the experimental design, could improve the comprehension by the readers. I suggest adding a new explanatory figure.
The abstract is poorly written. The abstract should include the methods and experiments clearly and shortly.
The introduction is not comprehensive enough for this title and subject. Please remove general sentences between lines 54-60 and replace them with more specific terms from the literature survey and research gap.
Please elaborate specifically the methods for each application for the aforementioned lines. Also, do the same for lines 82-89 and 94-98. Also in the introduction.
Please use and cite more recent papers about thermoresponsive hydrogels and potential applications; an example is the following:
--Blacklow, S. O., Li, J., Freedman, B. R., Zeidi, M., Chen, C., & Mooney, D. J. (2019). Bioinspired mechanically active adhesive dressings to accelerate wound closure. Science advances, 5(7), eaaw3963.
The mechanical analysis is absent in this manuscript. Fracture toughness is one of the important properties that could be measured by AFM/indentation or tensile tests (For all hydrogels). Please take a look at the below paper:
--Zhao, X. (2014). Multi-scale multi-mechanism design of tough hydrogels: building dissipation into stretchy networks. Soft matter, 10(5), 672-687.
The effect of affinity binding and solute diffusivity is absent from the current manuscript. It will be beneficial if you could add related equations from the literature. It will also be beneficial if you discuss about the relation between the affinity binding and diffusivity.
--Fatin-Rouge, N., Milon, A., Buffle, J., Goulet, R. R., & Tessier, A. (2003). Diffusion and partitioning of solutes in agarose hydrogels: the relative influence of electrostatic and specific interactions. The Journal of Physical Chemistry B, 107(44), 12126-12137.
General comment:
The quality of figures 2.3, 2.9, and 2.10 is not adequate. Please reproduce them with better quality and increase the font size for both axes.
Also for table 2.2, please, add a "vector format" table (like table 2.3) instead of a screenshot.
Author Response
Reviewer 1:
A more comprehensive graphical abstract, or an additional figure which explains the experimental design, could improve the comprehension by the readers. I suggest adding a new explanatory figure.
An explanatory chart for synthesis procedure of injectable pH and thermosensitive hydrogels have been added in Figure 4.1. as well as ex-ovo CAM assay method has been carried have been illustrated (Figure 4.2.). In addition, synthesis methods were explained more clearly for better understanding of the techniques in the revised draft.
The abstract is poorly written. The abstract should include the methods and experiments clearly and shortly.
The abstract has been updated by specifically explaining the methods and results.
The introduction is not comprehensive enough for this title and subject. Please remove general sentences between lines 54-60 and replace them with more specific terms from the literature survey and research gap.The sentences in lines of [54-60] has been removed and changed with more specific literature examples from the literature in details. Each application has been explained in more details and relevant references have been cited.
Please elaborate specifically the methods for each application for the aforementioned lines. Also, do the same for lines 82-89 and 94-98.
The mentioned lines dealing with use of different neutralising agents, have been explained with more details for each application in revised draft.
Also in the introduction. Please use and cite more recent papers about thermo responsive hydrogels and potential applications; an example is the following:--Blacklow, S. O., Li, J., Freedman, B. R., Zeidi, M., Chen, C., & Mooney, D. J. (2019). Bioinspired mechanically active adhesive dressings to accelerate wound closure. Science advances, 5(7), eaaw3963.
Different Thermosentive hydrogels and their applications have been added into the last manuscript and recommended paper and other very recent papers have been cited.
The mechanical analysis is absent in this manuscript. Fracture toughness is one of the important properties that could be measured by AFM/indentation or tensile tests (For all hydrogels). Please take a look at the below paper:
--Zhao, X. (2014). Multi-scale multi-mechanism design of tough hydrogels: building dissipation into stretchy networks. Soft matter, 10(5), 672-687.
With due respect to the reviewer, yes, mechanical properties analysis is very important. In the primary studies we have involved Rheology tests showing the
viscoelastic behaviours of hydrogels presenting the elastic modulus which is one mechanical property measurement. The further fracture toughness analyses have planned to be performed and present in a future paper.
The effect of affinity binding and solute diffusivity is absent from the current manuscript. It will be beneficial if you could add related equations from the literature. It will also be beneficial if you discuss about the relation between the affinity binding and diffusivity. --Fatin-Rouge, N., Milon, A., Buffle, J., Goulet, R. R., & Tessier, A. (2003). Diffusion and partitioning of solutes in agarose hydrogels: the relative influence of electrostatic and specific interactions. The Journal of Physical Chemistry B, 107(44), 12126-12137.
Good suggestion, indeed the affinity binding, and particle diffusion in counter charged systems dealing with pH sensitive drug delivery mechanism, and also oppositely charged polycomplex systems have been addressed which are related to present research. The suggested article has been cited in the current revised manuscript.
General comment:
The quality of figures 2.3, 2.9, and 2.10 is not adequate. Please reproduce them with better quality and increase the font size for both axes.
The figures (Figure 2.3, 2.9, and 2.10) have been reproduced and changed with the previous ones
lso for table 2.2, please, add a "vector format" table (like table 2.3) instead of a screenshot.
Table 2.1. and Table 2.2. which were in the screenshot format have been replaced with the tables in the “vector format”.
Reviewer 2 Report
The manuscript assessed “In-situ forming thermosensitive injectable hydrogels to stimulate angiogenesis: Potential candidates for fast bone regeneration applications” This research is under the scope of this journal; the topic is relevant for readers and this research deals with potentially significant knowledge to the field.
However, there are numerous issues in the present manuscript that need to be addressed before publication and English native review is strongly advisable.
- The Chitosan was also described for pulp regeneration, it could be mentioned in your introduction. To support your idea in line 58, please read the paper: Histologic Evaluation of Regenerative Endodontic Procedures with the Use of Chitosan Scaffolds in Immature Dog Teeth with Apical Periodontitis. Palma, PJ. et al. Journal of Endodontics, Volume 43, Issue 8 , 1279 – 1287.
- When mentioning materials or devices: for some of them you don't mention the manufacturer at all, for some you mention only the manufacturer, for some the manufacturer and city, for some you mention the manufacturer and city/ country.
- “ml” change for “mL”
- The references are not well inserted in the manuscript, and standardized:
line 507 - [68][69] must change for [68,69]
Author Response
Reviewer 2:
The manuscript assessed “In-situ forming thermosensitive injectable hydrogels to stimulate angiogenesis: Potential candidates for fast bone regeneration applications” This research is under the scope of this journal; the topic is relevant for readers and this research deals with potentially significant knowledge to the field. However, there are numerous issues in the present manuscript that need to be addressed before publication and English native review is strongly advisable.
We have combed through the manuscript and the English language has been improved.
The Chitosan was also described for pulp regeneration, it could be mentioned in your introduction. To support your idea in line 58, please read the paper: Histologic Evaluation of Regenerative Endodontic Procedures with the Use of Chitosan Scaffolds in Immature Dog Teeth with Apical Periodontitis. Palma, PJ. et al. Journal of Endodontics, Volume 43, Issue 8, 1279 – 1287.
The suggested paper has been cited, and another clinical example in the related dental application of chitosan has been also added in the revised manuscript.
When mentioning materials or devices: for some of them you don't mention the manufacturer at all, for some you mention only the manufacturer, for some the manufacturer and city, for some you mention the manufacturer and city/ country.
Manufacturer information for each device have been revised and added accompanying with their origin including city and country.
“ml” change for “mL”
“ml” has been changed for “mL” as unit
The references are not well inserted in the manuscript, and standardized:
line 507 - [68][69] must change for [68,69]
The given example reference style [68,69] has been applied for [68][69] and corrected all other citations as well in the revised draft.
Round 2
Reviewer 1 Report
Dear authors,
[1] I asked you to add more about affinity binding and salute diffusivity which are absent yet.
[2] Also for the introduction, you should cover a wider range of the different thermoresponsive gels and different application as well.
Author Response
Dear Editor
First of all, my sincere apologies for submitting the wrong version of the manuscript in the previous round.
We are again thankful to both the reviewers for their valuable comments which have been extremely useful in modifying this manuscript.
In this cover letter, the changes incorporated in the manuscript are outlined and the revised version of the manuscript with tracked changes is separately attached.
We have addressed all comments or provided a suitable rebuttal, and we believe that the revised manuscript is significantly improved and very much hope that it will be acceptable for publication in IJMS now.
In the second round the manuscript has been further modified and points raised have been elaborated.
Round 2 comments are given at the end of modification details of Round 1.
I have combed through the paper and have made significant changes throughout the manuscript. For clarity both the previous and the latest modified versions are attached for consideration.
If you may require any further information, please do not hesitate to contact me.
Yours sincerely
Prof Ihtesham ur Rehman

Round 3
Reviewer 1 Report
Dear authors,
Thanks for the second revision.